# Explanation-guided dynamic feature selection for medical risk prediction

Nicasia Beebe-Wang [1]   Wei Qiu [1]   Su-In Lee [1]

## Abstract

In medical risk prediction scenarios, machine learning methods have demonstrated an ability to learn complex and predictive relationships among rich feature sets. However, in practice, when faced with new patients, we may not have access all information expected by a trained risk model. We propose a framework to simultaneously provide flexible risk estimates for samples with missing features, as well as context-dependent feature recommendations to identify what piece of information may be most valuable to collect next. Our approach uses a fixed prediction model, a local feature explainer, and ensembles of imputed samples to generate risk prediction intervals and feature recommendations. Applied to a myocardial infarction risk prediction task in the UK Biobank dataset, we find that our approach can more efficiently predict risk of a heart attack with fewer observed features than traditional fixed imputation and global feature selection methods.

## 1. Introduction

In many medical decision-making scenarios, there is a trade-off between collecting more features at a cost, versus making a decision about a patient based on what is currently known. In general, observing more information about the person can lead to more accurate and confident predictions, and current machine learning (ML) prediction models often expect a rich feature set, which may not be available at all times in practice. To that end, previous work has focused on designing models that can efficiently choose features dynamically, tailored to the context of a specific sample; however, they often rely on specific modeling/architecture innovations (e.g, decision trees (Xu et al., 2014; Viola & Jones, 2004)). Another line of research has involved sensitivity-

---

[1]Paul G. Allen School of Computer Science & Engineering, University of Washington. Correspondence to: Su-In Lee <suin-lee@cs.washington.edu>.

*Workshop on Interpretable ML in Healthcare at International Conference on Machine Learning (ICML)*, Honolulu, Hawaii, USA. 2023. Copyright 2023 by the author(s).

based approaches that aim to select features based on the unknown features' influence on model predictions (Early et al., 2016b;a; Kachuee et al., 2019); however, these approaches have also tended rely on a specific class of model (linear models or autoencoders (Ma et al., 2019)). For a fixed supervised ML model (about which we make minimal assumptions), we propose a general framework to simultaneously provide both flexible risk estimates for individuals with missing features and personalized feature recommendations to identify which missing features may be most informative to select next given the context.

Our approach (Figure 1) relies on three main components: (1) a conditional feature imputer for sampling possible values of missing features given the observed ones, (2) a fixed supervised ML model, and (3) local feature explainer for the model. Empirically, we find that an approach using KNN-based imputation, an XGBoost model, and a SHAP explainer is able to more efficiently predict 10-year risk for myocardial infarction than traditional single value imputation and fixed (global) feature selection.

## 2. Methods

For our approach, given an individual with partially observed features (e.g., a sparsely populated medical record), we stochastically impute their missing features–ideally, drawing from the conditional distribution of their missing features given observed ones–to generate examples of a complete record given the currently available information. We then use this "ensemble" of imputations to: (1) provide a risk interval (rather than just a single point estimate) around the adverse event, and (2) select a feature to collect next, guided by model explanations, which may best help reduce uncertainty with respect to the model's prediction. Such an approach may enable clinicians to make better informed decisions by providing them with flexible risk intervals (regardless of how much information is already known about the patient) and suggesting follow-up tests to improve their understanding of a patient's risk profile. Our approach is model agnostic, and relies on three main components:

**(1)** A conditional feature imputer for generating ensembles of imputed samples. Our goal for the stochastic imputer is to conditionally sample missing features given observed ones such that, by sampling an "ensemble" of multiple im-

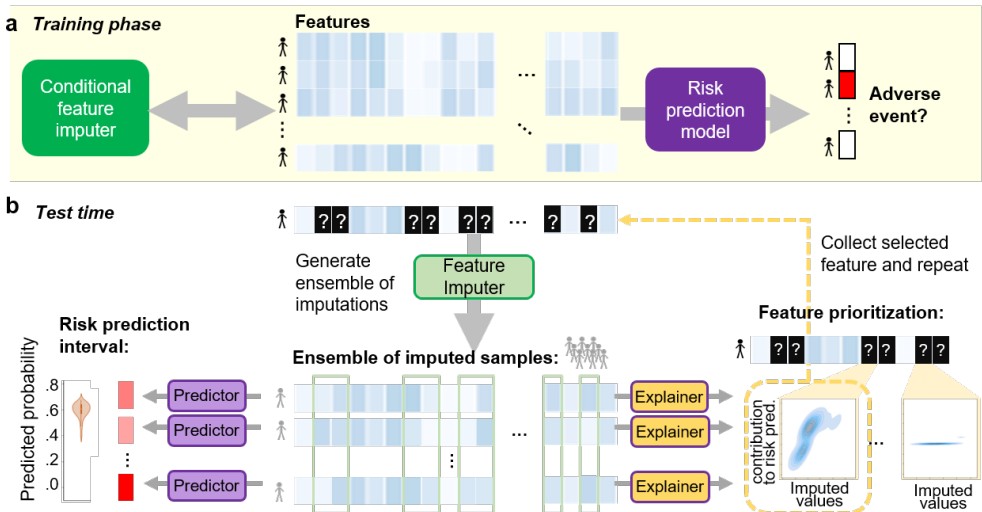

*Figure 1.* Overview of our approach. (a) Our method relies on a fixed conditional feature imputer and risk prediction model fit to a training set. (b) For each incomplete sample at test time, we generate an ensemble of imputed samples and use the prediction model to produce a risk interval (left). The next feature is dynamically selected based on model explanations, and if observed, the process may be repeated.

putations for a given individual, we obtain a distribution of possible complete samples for that person. These imputation ensembles could then be fed into the predictor and explainer to generate a distribution of predicted risk scores and explanations conditioned on the observed context. In practice, developing generative models for conditional imputation is challenging, and an area of active research (e.g., neural network methods such as GAIN(Yoon et al., 2018) and GI(Kachuee et al., 2020)); however, we find that a simple approach of sampling nearest neighbors works well in practice (see Results). **(2)** A supervised ML model. For our approach, we assume that the model has already been trained on a labeled dataset consisting of a fully observed feature set and the clinical label of interest. We make minimal assumptions about the type of model used (for our experiments, we use an XGBoost (Chen & Guestrin, 2016) model), and only require that the model has an associated **(3)** feature explainer, which can estimate each feature's contribution to the model's prediction for a given sample (for our experiments, we use SHAP (Lundberg & Lee, 2017)).

For our experiments, we used a fixed training set for all components. However, we note that because the stochastic imputer and prediction model are trained independently, their training sets do not need to be identical. This may be particularly advantageous if there are many more unlabeled samples available (which may be used to train the imputer).

### 2.1. Putting it all together: Our approach

At test time (Figure 1b), from the imputation ensemble generated by the conditional feature imputer, we simply use

the risk model's predictions to generate a risk distribution (whose spread reflects variation in the model's output with respect to imputed features). We further propose using the distributions of feature attributions to inform feature recommendations. In particular, we hypothesize that using variance in SHAP values across the imputation ensemble will be an effective metric for selecting the next feature. Intuitively, large variations in SHAP values would indicate that the model's predictions are sensitive to our simulated variations in the missing features. In contrast, if a feature has high variance in the imputed values but not SHAP values, that would indicate that these variations are not relevant–according to the prediction model–to risk. Similarly, a feature with high-magnitude but low-variance SHAP values may also be a poor choice, since given the current context, we are already confident about how the unknown feature would impact the model (for example, if we have two redundant features and have already observed one of them).

### 2.2. Experiments

We first evaluate our approach on a toy dataset with a known conditional distribution where we can compare feature selection approaches independently of imputation methods. We then apply our approach to data from the UK Biobank (www.ukbiobank.ac.uk), a biomedical database containing data from individuals across the UK, including hundreds of features collected during an initial visit and detailed long-term health outcomes. For our analyses, we focus on a randomly selected subset of 100,000 samples, along with 252 features (described further in Appendix A.2), and we

observe that about 2% of individuals have a reported myocardial infarction within 10 years after their initial screening.

## 3. Results

In Appendix A.1, we show the advantages of our dynamic feature selection approach on a synthetic dataset where we directly impute missing features from a known conditional distribution, and thus can evaluate our feature selection approach in isolation. As shown in Appendix Figure A.4, compared with a fixed global ordering, our explanation-guided dynamic feature selection strategy more efficiently identifies relevant features given context from observed ones.

We now turn to the task of 10-year myocardial infarction prediction in the UK Biobank (UKB) dataset. We first fit a supervised ML model and imputer to our training set. We then use a test set to simulate an interactive process in which we alternate between (1) generating a prediction given the current observed information, and (2) selecting the next feature to un-mask. We then observe how the prediction model's average performance and prediction variation progress as features are selected and observed.

### 3.1. Initial model training

***Supervised risk model and explainer.*** For the 10-year myocardial infarction prediction task, we fit an XGBoost model to the training set to establish baseline performance expectations for a complete feature set, for which we obtain test AUROC and AP scores of 0.768 and 0.088, respectively (Appendix Figure A.5). We use SHAP as our model's explainer, and note that the top 20 features in the training set ordered by mean absolute SHAP value (listed in Figure 3b and shown in Appendix Figure A.5c) cover a range of feature types, from demographic features to medical lab tests.

***Conditional feature imputation.*** As described in Methods, our approach relies on a conditional feature imputer which samples missing features conditioned on the observed ones. We empirically find that a k-nearest neighbors (KNN)-based approach, with an ensemble size of 100, works well in practice (Appendix A.3). Across different rates of induced missingness, we find that this approach leads to the most accurate risk predictions (averaged across our XGBoost model's outputs for the ensemble of imputed samples) compared with standard approaches such as mean-value imputation and imputing from the marginal distribution, or recently proposed deep learning imputation models (Kachuee et al., 2020; Yoon et al., 2018) (Appendix Figure A.5b).

### 3.2. Iterative feature selection experiments

We now demonstrate the value of our approach for iteratively predicting risk intervals and dynamically selecting

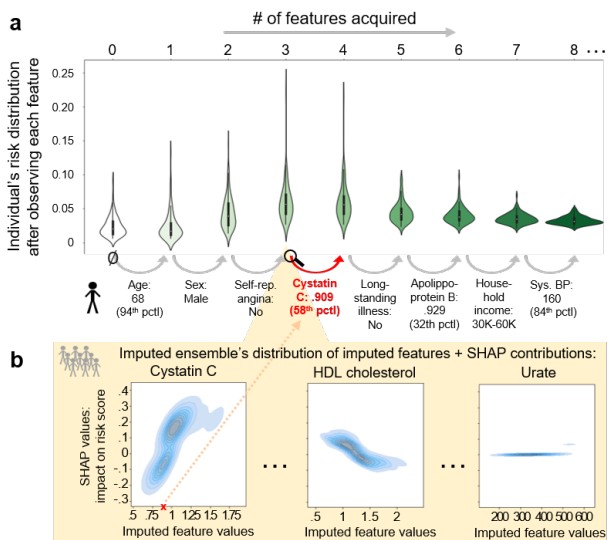

*Figure 2.* Example of our approach for a single sample. (a) Distribution of risk scores from imputation ensembles after observing each feature. (b) At a specific time-point, examples of imputed values vs. resulting SHAP values for candidate features. Cystatin C is selected because it has the highest variance in SHAP value.

features. For each test sample, we run the following procedure (illustrated in Figure 2) beginning with 0 observed features, until 20 features have been observed: (1) Generate an ensemble of 100 imputed samples using the KNN-based approach described above, (2) use the supervised risk model to generate predictions for each of the 100 imputed samples, resulting in a predicted "risk interval", (3) apply SHAP to obtain feature contribution scores for each feature across each imputed sample, and finally, (4) choose the feature with the highest SHAP value variance and uncover the true value of that feature.

To illustrate the potential use of our approach in practice, we show an example for a single participant in the UKB study in Figure 2. In Figure 2a, we first demonstrate the individual's prediction interval evolves as each new feature is collected (resulting in a new ensemble of imputations, and subsequent risk scores). As more features are observed, we see that the prediction intervals tend to shrink indicating that uncertainty of the model predictions with respect to missing features' imputations is decreasing. In Figure 2b, we show an example of imputed values vs. their contribution to the model's risk scores (according to SHAP values) after observing three initial features. In this particular example, imputed values for Cystatin C have the highest-varying impact on model predictions given the context at that point, and is thus selected next. Our approach, as highlighted in this example, provides an explanation-guided recommenda-

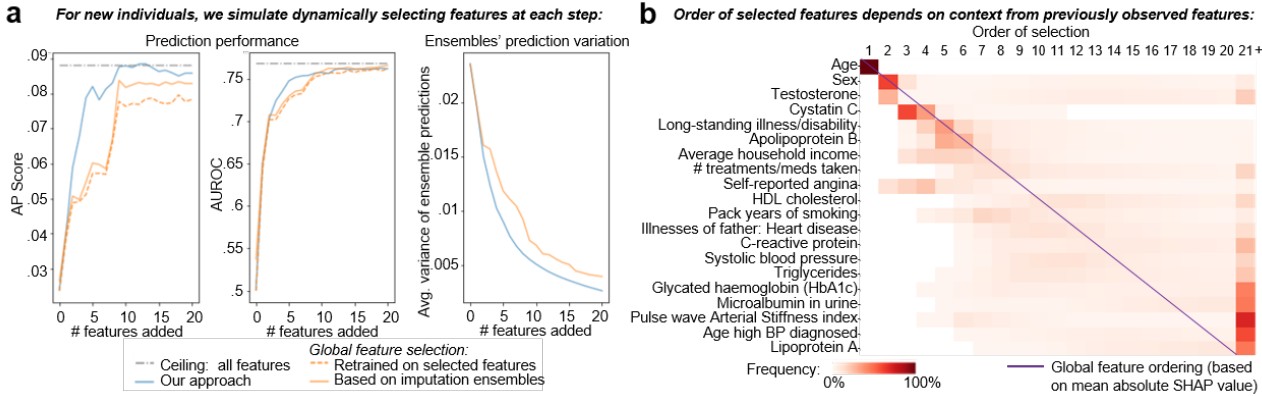

*Figure 3.* (a) Comparison between our dynamic feature selection approach and global feature selection for 10-year myocardial infarction prediction in the UKB dataset. (b) Overview of our feature selection orders compared with fixed global feature selection (in y-axis).

tion (based on ensembles of imputations) for features, and thus may provide users with context-dependent insight into why a feature may be useful to observe.

In Figure 3a, we report the performance of our approach, aggregated across the test set, as features are iteratively selected and observed. For comparison, we consider two baselines using fixed feature orderings provided by our global feature ranking (based on mean absolute SHAP value; Appendix Figure A.5c). First, we consider the same procedure described above, but replace the dynamic feature selection strategy (steps 3 and 4) with simply choosing features in the fixed global order. Second, a more traditional approach of using global feature selection involves re-training the prediction model on the selected features. Thus, we also consider a collection of models trained with feature budget (i.e., a single-feature model trained on age, a two-feature model trained on age and sex, ..., and a 20-feature model trained on the top 20 features), as such a model is tailored to the specific expected use-case.

From our experiments, we find that our approach more efficiently prioritizes features than fixed global feature selection strategies, leading to improved prediction accuracy with fewer features observed (Figure 3a). In Figure 3b, we show that our approach does indeed lead to different feature orderings tailored to samples' context of previously observed features. We see that age is always selected first, but that the participant's age informs the choice of the second feature, whose value informs the next feature choice, and so on. Appendix Figure A.6 provides a detailed view of feature orderings across the test set. Anecdotally, we note that for younger individuals, our approach tends to choose testosterone next; sex is often not selected till much later (perhaps because sex can be easily inferred from the testosterone values and thus would provide redundant information).

## 4. Discussion

In this work, we propose an approach to leverage conditional imputation and explainability methods to provide flexible health risk estimates and context-dependent feature recommendations for a fixed ML prediction model. Applied to a 10-year myocardial infarction prediction task in the UKB dataset, our approach (implemented with KNN-based imputation, XGBoost, and SHAP) led to more efficient feature prioritization compared with a static approach and single-value imputation. One key consideration is that our method relies on having a reliable way to conditionally impute missing features. While our KNN-based approach worked well in practice, it is not guaranteed to be effective, and may be particularly limited in the case of small datasets where neighborhoods may be particularly sparse.

There are several natural extensions of our work to consider. First, our feature selection strategy does not take into account the fact that those features tend to have varying costs. One simple extension is to adjust our feature collection policy to balance our current metric (i.e., SHAP variance) with the cost of the feature. Second, our experiments considered collecting features over a fixed budget of 20 features. In practice, it may be valuable to leverage some notion of uncertainty (contained in risk prediction intervals) in a triage setting, where we may allocate a feature budget unevenly across samples (e.g., terminating a sample's collection procedure once its risk interval is sufficiently narrow).

Although further experimentation is needed to validate our method in a real-world setting alongside state-of-art methods, initial experiments demonstrate its potential to provide flexible risk estimates despite incomplete medical information, along with context-dependent feature recommendations which may aid clinicians in seeking additional clarity.

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

# A. Appendix

## A.1. Toy dataset experiment

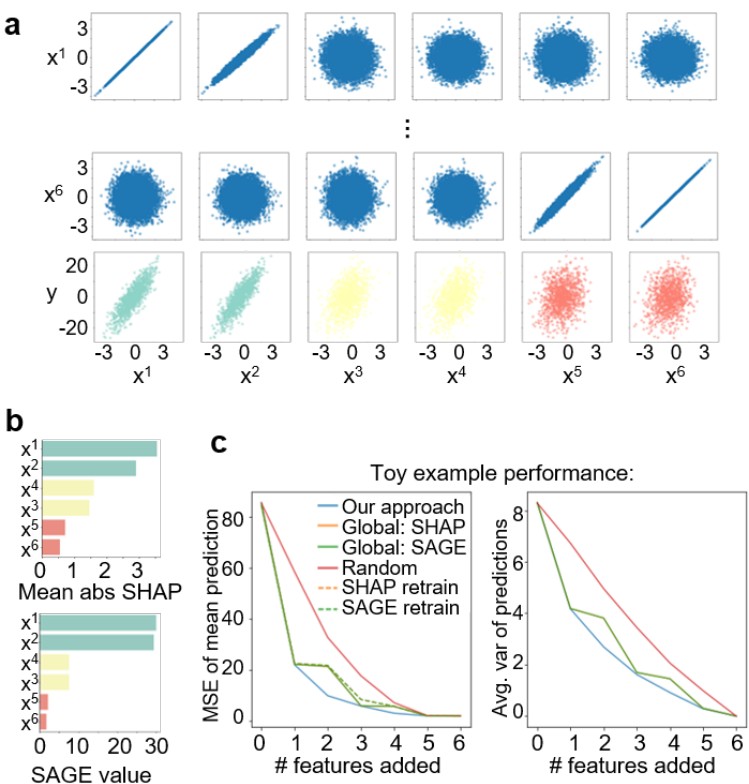

*Figure A.4.* A toy example illustrating the advantage of our method of dynamic feature selection over a fixed ordering. (a) Scatter plots illustrating the relationships among features and a target variable in a simulated toy dataset. The feature set contains 3 pairs of correlated features which are independent of other pairs. Our target feature is a function of all features, but relies most heavily on $x^1$ and $x^2$ and least on $x^5$ and $x^6$. (b) Global feature importances for an XGBoost model trained to predict $y$ on the simulated data shown in a. To compute global feature importance, we consider both SHAP (where we compute the mean absolute SHAP value)(Lundberg & Lee, 2017) and SAGE(Covert et al., 2020). (c) When simulating an interactive risk prediction and dynamic feature selection process on synthetic data illustrated in a, we assess the average performance of our approach on samples (y-axis) when starting at 0 information and then selecting a new feature at each time point (x-axis) with respect to both error (left) and ensembles' prediction variation (right). Baselines: "random" follows our multiple imputation and prediction procedure, but with the next feature selected randomly at each time point. "Global: SHAP" and "Global: SAGE" follow our multiple imputation and prediction procedure, but features are acquired in order of global feature importance as shown in b. "SHAP retrain" and "SAGE retrain" use the same global feature orderings from b, but we retrain models at each feature budget.

### A.1.1. DATA GENERATING PROCESS, AND CONDITIONAL IMPUTATION

We first consider an example with synthetic data where we can stochastically impute samples directly from a known conditional distribution (thereby testing the dynamic feature selection model independently of the performance of an imputer). As illustrated in Figure A.4, the toy dataset contains pairs of correlated features, and a prediction task $y = 4(x^1 + x^2 + \epsilon) + 2 * (x^3 + x^4 + \epsilon) + (x^4 + x^6 + \epsilon)$ where $\epsilon \sim N(0, 0.01)$ represents random noise. The six features are normally distributed (mean 0, unit variance), and consist of three pairs of redundant features that are independent of the other pairs ($\rho = .9$ between paired features, as illustrated in Figure A.4a). Thus, we have simulated a case in which we have pairs of redundant features, where within each pair, the features contribute equally to $y$, and there's a clear ranking of importance between pairs.

Given this data generating process, it's then straightforward to randomly sample imputations from a partially observed sample: for each correlated pair of features $x^a$ and $x^b$, if $x^a$ is missing but $x^b$ was observed, we sample from the conditional distribution $x^a | x^b \sim N(.9 * x^b, .19)$ (or vice versa). Otherwise, if both are missing for a given test sample, we randomly sample from the marginal distribution $x^b \sim N(0, 1)$, and then sample $x^a$ conditioned on $x^b$ as just described. Given a dataset simulated with this data generating process, we can now mask features and use our approach above to directly sample imputations for the masked features conditioned on the observed ones.

A.1.2. FEATURE ACQUISITION EXPERIMENT

For the supervised prediction model, we train an XGBoost model (with default parameters) on a 10,000-sample simulated training set. First, we apply SHAP (a local feature attribution method) (Lundberg & Lee, 2017) and SAGE (a global feature attribution) (Covert et al., 2020) method to assess how an XGBoost predictor model relies on our simulated features to generate predictions. Consistent with the data generation process, these methods both identify a clear ranking among the pairs of features (Figure 3b), although the relative importance within pairs slightly differs. In a standard global feature acquisition strategy, features would always be selected based on this global ranking (i.e., $x^1 \to x^2 \to x^4 \to x^3 \to x^5 \to x^6$).

We now turn to the evaluation of our approach on this toy example. In Figure A.4c, we show the average performance of our approach when simulating prediction and selection of features from no observed features until all features have been observed. In particular, for each of 1,000 test samples, we repeat the following steps until all features have been observed: (1) we sample 100 imputed samples conditioned on the features observed up until now, (2) we report the mean and variance of the model predictions on these 100 imputations, (3) we then compute SHAP values for each of the imputed examples and select the missing feature with the highest-variance SHAP value, (4) we un-mask the missing feature. In Figure A.4c, we also show how global feature selection policies compare with our context-aware approach (while still using stochastic imputation to generate prediction intervals). We additionally show results for "retrained" global selection approaches because in practice, when using global feature selection, given that the order is fixed, it is reasonable to re-train a prediction model with the subsets of features that would be used at test time (i.e., a 1-feature model containing the top feature, a 2-feature model containing the top two features, etc.). In this toy example, we find that our dynamic feature selection approach allows our method to more efficiently gather features at test time. By modeling the missing features conditioned on the observed ones, our approach avoids collecting redundant features that would be unlikely to substantially alter the model predictions, unlike global feature selection policies (Figure A.4c).

## A.2. UK Biobank (UKB) data overview and preprocessing

UKB participants were enrolled between 2007-2014 from 21 assessment centers across England, Wales, and Scotland. Our study includes all measurements taken during their initial visit, available on December 13th, 2021. During an initial comprehensive visit, hundreds of features were collected, including information about sociodemographic and lifestyle factors, health and medical history, cognitive testing, physical measures (such as composition and hearing tests), and lab tests from biological samples (including blood and urine). We exclude (1) features that are missing in more than 80% of the samples, and (2) highly correlated features with correlations greater than 0.98 (when such correlations existed, we kept just one of the features and removed the others). After excluding features, our UKB dataset has 825 features from numerous categories: demographics, blood assays, health and medical history, lifestyle and environment, physical measures, etc.

For our analyses, we initially considered seven health outcomes which are provided by the UKB database as "algorithmically-defined outcomes," meaning that they are outcomes linked to hospital admissions and death registries (`https://biobank.ndph.ox.ac.uk/showcase/label.cgi?id=42`): chronic obstructive pulmonary disease, asthma, all-cause dementia, end-stage renal disease, myocardial infarction, all-cause parkinsonism, and stroke. One of the most densely annotated outcomes, and the focus of our analyses, are myocardial infarctions (also known as heart attacks; around 2% incidence over 10 years). For our analyses, we use a ten-year follow-up period as our prediction goal. For each of these conditions, we considered the individual to be a control case if they had no history of the condition during their intake visit as well as no report of the condition within 10 years after the visit, and a positive case if they had no prior history of the condition at their intake and subsequently had a reported incidence of the condition within the next ten years. For a given condition, the label was considered to be unknown if they had a pre-existing report of the condition during their intake, or if they had a report with an unknown time, and such samples were excluded from training our supervised models.

For training and evaluating our models, we used a randomly selected sample of 100,000 individuals, which we divided into training (64,000 individuals), validation (20,000), and test (16,000) splits. In order to train our models, we performed

the following additional preprocessing steps: (1) We imputed missing features using MissForest (Stekhoven & Bühlmann, 2012), a nonparametric random forest-based multiple imputation method for mixed-type data, (2) we normalized features to have 0 mean and unit-variance (which was necessary to compute distances for KNN-based imputation with similarly scaled dimensions).

Finally, we generated a reduced feature set for our final models: for each of the seven conditions listed above, we trained a separate XGBoost model to predict whether the condition occurred within 10 years based on the full 825 feature set. We then applied SHAP to each of the seven models and found that many of the features played no significant role among any of the models (SHAP values of 0 across all samples). Thus, to provide a more reasonable starting point for our models, we chose a reduced feature set of 252 features consisting of all features that had a mean absolute SHAP value of at least 0.001 for at least one of seven outcome models. Our final data for the myocardial infarction risk prediction experiments consisted of 62,444 training samples (from the initial 64,000 training samples, we excluded samples with unknown labels) with the 252 features described above, and 15,612 test samples which were used to evaluate the effectiveness of our approach. In our experiments, the 62,444-sample training set is used for both fitting the XGBoost model, as well as the KNN-based imputation approach.

### A.3. UK Biobank experiment details

For our feature imputer, we use KNN-based imputation with an ensemble size of 100. In particular, we use an imputation ensemble size of 100, meaning that for a given partially observed test sample, we identify the 100 nearest neighbors in the training set (with distances computed in Euclidian space for normalized features based on the observed features only). We then generate an ensemble of imputations where, for each sample in the ensemble, the observed features are kept as is, and the remaining unobserved features are imputed from the record of the selected neighbor. For our XGBoost risk prediction model, we used the implementation from Chen & Guestrin (2016) and used default hyperparameter settings. Once fit, this trained model was used as the fixed supervised model across all feature collection strategies.

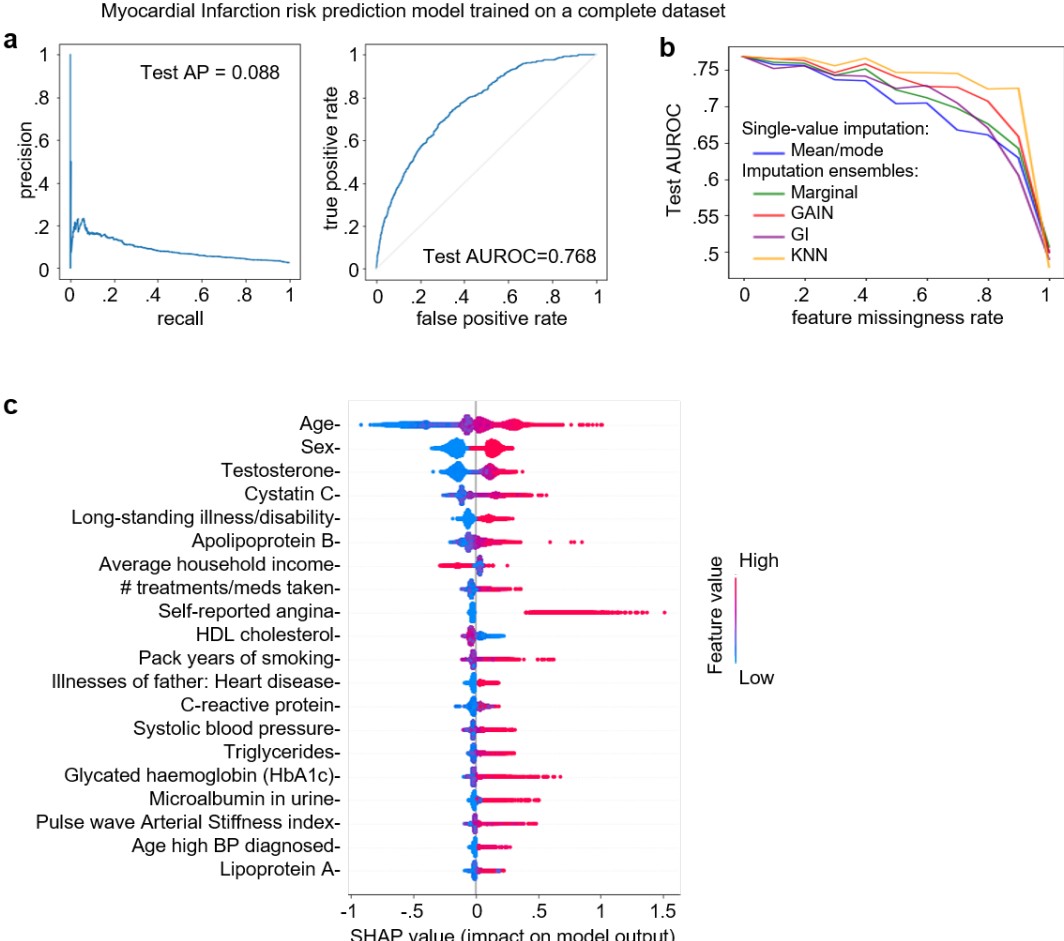

*Figure A.5.* Summary of our 10-year myocardial infarction XGBoost model. (a) Test set performance when considering the full feature set without missingness. (b) Simulating MCAR missingness at different rates, we compare the effectiveness of different imputation strategies. For each strategy (except mean/mode), we generate an imputation ensemble for each sample (100 imputed samples for each real sample) and consider the average of model predictions across the ensemble as our final risk estimate. For mean/mode imputation, we simply impute missing features with the mean (for continuous features) or mode (for binary features). We also evaluate the following strategies: marginal (uniformly sampling the feature's value across the entire training set), GAIN (a neural network-based approach by Yoon et al. (2018)), GI (a neural network based approach by Kachuee et al. (2020)), and KNN, our final selected approach of identifying nearest neighbors in the training set based on observed features and directly using features observed in those neighbors. (c) SHAP summary plot for the trained model: distribution of training set values vs. their impact on the model output (SHAP values). The features are sorted by mean absolute SHAP value, and this ranking is used as our feature ordering for the fixed feature selection strategy.

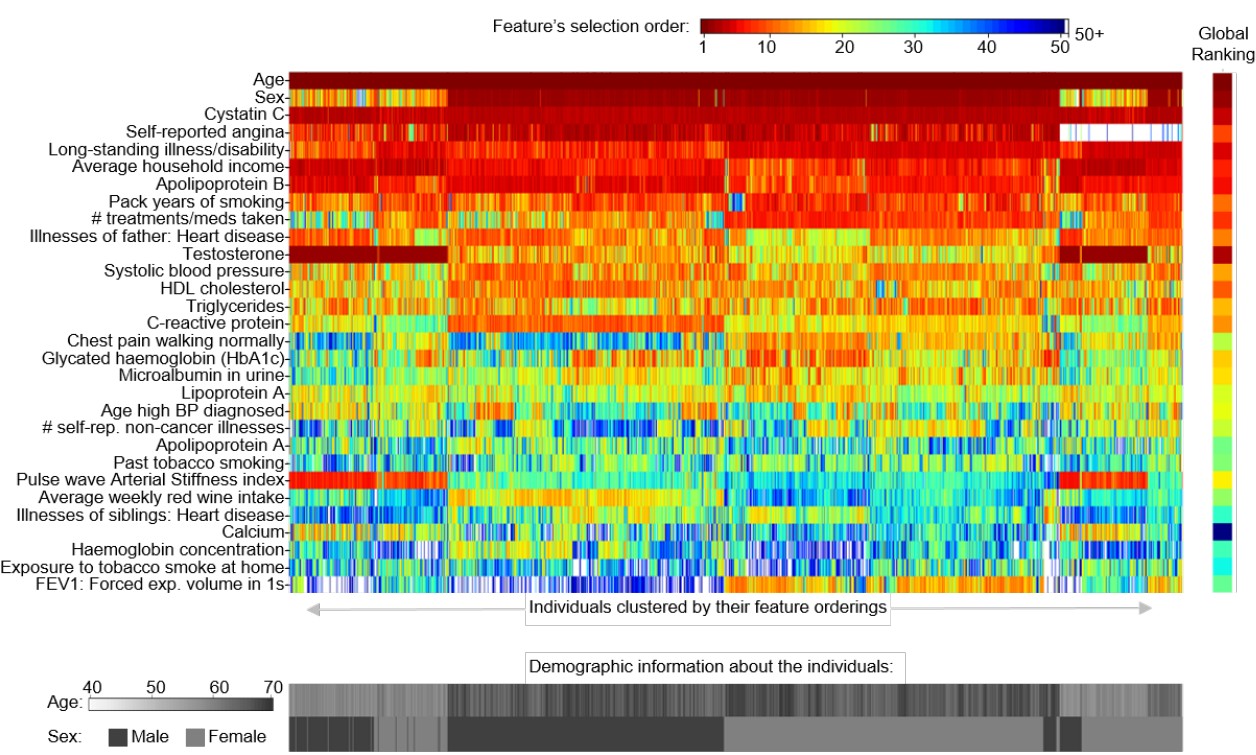

*Figure A.6.* Feature orderings across all samples in the test set. In the top heat map, each column represents a sample in the test set, and the cell color indicates the order in which features were collected (dark red for a sample's first selected feature, dark blue for the 50th, and white if the feature was selected after the top 50 features for that sample). Below the main heatmap, we also show each sample's age and sex for to highlight some possible relationships between collected features and the subsequent ordering of later features (e.g., for younger subjects, our approach tends to select testosterone early instead of sex). To the right, we also show the global ranking of features which is used for the fixed global feature ordering baseline.

