# OpenReview forum: "Explanation-guided dynamic feature selection for medical risk prediction"
_ICML.cc/2023/Workshop/IMLH — IMLH 2023 Poster_

### Official Review · Reviewer_1q42 · 2023-06-11
**Comments on IMLH 2023, Submission 92**

**Rating:** 7
**Confidence:** 4

**Review:**

In this study, a framework is proposed to address the simultaneous provision of flexible risk estimates for samples with missing features and context-dependent feature recommendations to identify the most valuable information to collect next. The proposed approach utilizes a fixed prediction model, a local feature explainer, and ensembles of imputed samples to generate risk prediction intervals and feature recommendations. The effectiveness of the method was evaluated using two datasets, and the results substantiate its efficacy.

However, to underscore the advantages of the proposed method, it is recommended that the author compare it with state-of-the-art models. This comparison would enable a demonstration of the prediction performance and emphasize the flexible risk estimation capability of the proposed method.

---

### Official Review · Reviewer_LqKp · 2023-06-18
**Interesting paper to address missing feature problem**

**Rating:** 6
**Confidence:** 3

**Review:**

This paper proposes a model-agnostic framework to provide both flexible risk estimates for individuals with missing features and personalized feature recommendations to identify which missing features may be most informative to select next given the context. It first uses a conditional feature imputer to generate possible imputations, then estimate a risk distribution, and finally select features based on their impacts on the risk. This method is interesting and the authors provide solid experiments and analyses to validate its effectiveness.

---

### Meta-Review · Area_Chair_JA5k · 2023-06-19

**Recommendation:** Accept (Poster)
**Confidence:** 4

**Metareview:**

Reviewers are generally positive in recommending the acceptance of this manuscript but also raise concerns. Please address them in the final version.

---

### Decision · Program_Chairs · 2023-06-20

Accept (Poster)